# Psychological and Neurophysiological Screening Investigation of the Collective and Personal Stress Resilience [note 1]

**DOI:** 10.3390/bs13030258

**Published:** 2023-03-15

**Authors:** Sergey Lytaev

**Affiliations:** Department of Normal Physiology, St. Petersburg State Pediatric Medical University, 194100 Saint Petersburg, Russia; physiology@gpmu.org; Tel.: +7-812-416-5226

**Keywords:** stressors, time deficit, cognitive functions, activation factors, EEG, psychological testing

## Abstract

Methodological approaches to assess the human cognitive status are constantly evolving. At the same time, the creation of new assessment methods is accompanied by traditional research. This paper discusses the direction of research on the search for markers of stress resilience. The basis for the formation of the research algorithm was the assessment of activation factors of emotional states, including preceding stress–sensory (cognitive and informational) and psycho-emotional factors. This was determined using methodological techniques, stress factors, working conditions in professional teams, etc. For an express analysis (25–40 min) of diagnosing stress resistance, a research algorithm was justified, consisting of clinical and psychological testing, as well as EEG with traditional tests and analysis of indicators and spectra. Therefore, this research was aimed at the psychological and neurophysiological substantiation of approaches to express algorithms for assessing cognitive functions and resilience to stress under time deficit. A study on 102 healthy subjects and 38 outpatients of a neuropsychiatric clinic was performed. Basic outcomes: the integrative indicator SCL-90-R—”general index of severity” has a high statistical significance (*p* < 0.05) in both healthy subjects and neuropsychiatric outpatients. The effectiveness of the Mini-Mult test in conditions of time deficit is determined by the results of the scales of hypochondria, depression, hysteria, paranoia, psychasthenia, schizoid and hypomania (*p* < 0.05). Furthermore, we used a line of logical thinking techniques. A line of four logical methods is highly informative in assessing the mental status in conditions of time deficit. EEG power indices and spectra in theta, delta and alpha frequency ranges are an effective reflection of cognitive status. In this article, a testing algorithm as a variant for assessing neurocognitive status in screening studies of large groups is discussed.

## 1. Introduction

Today in both science and practical application, a stable set of methods has been formed to assess the complex human neurocognitive functioning. This includes both traditional psychological testing, as well as complex registration of a number of parameters—EEG, event-related potentials, ECG, EMG, fMRI, respiratory and vegetative parameters that characterize physical, mental (including cognitive) and professional health [1,2,3,4]. Recently, approaches utilizing the modeling of virtual reality and neural networks have been developed [5,6]. All of the above is undoubtedly of scientific interest. However, the use of such complexes takes time and the recruitment of profile specialists.

The physiological consequences of psychological stress as a form of interaction between a person and the environment depend on the person’s perception of his ability to cope with the stressor [7,8]. Human perception of stress involves the activation of three interconnected biological systems [9,10]. First, stress is perceived by sensory systems that evaluate and compare the stressful situation with the current state and previous stressful experiences of the organism. Second, the brain activates the autonomic nervous system through the sympathetic-adrenomedullary system and triggers a rapid release of catecholamine, noradrenaline and adrenaline. Thirdly, there is a simultaneous activation of the hypothalamic-pituitary-adrenal system, which leads to the release of adrenal glucocorticoids and cortisol in humans [8,9,11]. Thus, we perceive stress factors both directly through sensory systems and indirectly through chains of stress responses. Sensory systems are an information channel that is evaluated by physical and neurophysiological parameters [12,13]. Known vegetative chains of stress reactions, as a rule, are assessed using indirect parameters—vegetative, energy and hormonal, as well as by using psychological testing batteries [14,15].

Comparing information obtained from direct and indirect measurements to assess performance fluctuations presents the most significant methodological challenge [11,16,17]. A person who does excellent work on direct output indicators may experience noticeable fluctuations in indirect indicators. The price of such behavioral responses is excessively high, and the nature of the information obtained as a result of measurements can be misleading concerning a high level of performance [1,18].

Complex vegetative reactions to psychological stress, for example, during exam sessions for students, activate various endocrine and immune processes. Some of these mechanisms can be easily and non-invasively assessed by the content of a number of substances in saliva—cortisol, alpha-amylase and pro-inflammatory cytokines. It is known that glucocorticoids (cortisol) play a decisive role in reactions to stress [9]. Thus, stressful events (exam sessions, surgical interventions, dental procedures, etc.) are accompanied by an increase in cortisol levels. Such data can form the basis for the possible implementation of further anti-stress measures in practice and can lead to a better organization of stress resistance, which will allow, for example, students to study more efficiently or better cope with the requirements of university work [19].

It is believed that artificial intelligence (AI) technologies being created should improve social relations and progress in professional activities [6,18,20]. However, research shows that AI can create a number of ethical decision-making issues that contribute to the formation of stress. Authors refer to such problems, including algorithmic discrimination, underestimation of incoming information and, as a result, a decrease in the level of managerial responsibility [20,21]. Technological uncertainty, incomplete data and management errors are major sources of ethical risk in AI decision making. Thus, the adjustment of risk management elements can effectively block social risks arising from algorithmic, technological and information processing risks.

In particular, the risks associated with artificial intelligence technologies are proposed to be assessed with the application of the rooting theory, which is considered a common method in qualitative research. The essence of this theory is based on the facts of processing and conceptualization of unstructured data obtained from the collection of information and interviews [22]. There are seven stages in rooting theory: research goal, data collection and comparison, open coding, spindle coding, selective coding, theory saturation test and theory building [23]. Data collection and layered coding are the two most important steps in rooting theory. Using a three-level coding process, heterogeneous data can be generalized and then a standardized model can be built. Based on the theory of rooting, two main categories have been obtained—“technical risk identification” and “managerial risk identification”. Technical risk identification includes algorithm risk, data risk and technology risk, which occupies 36.5% of the first level nodes. Management risk identification includes both management risk and risk management [21,23].

In the last 20–25 years, due to changes in legislation in the implementation of medical activities, the organization and control of both the activities of medical specialists and the diagnostic measures they carry out have changed. In conditions where the criteria for evaluating medical activity are losing their differentiation and are based on the opinion of experts, sometimes heterogeneous and contradictory, in the absence of a specific legal framework on issues of “medical error” and “responsibility of doctors”, it becomes necessary to develop and formulate clear algorithms for diagnoses and treatment [24,25].

In this regard, it becomes a fundamental challenge for health systems to adapt to changes in the structures of health services that require technological and scientific innovations. The pace of multiple and interconnected challenges is placing additional strain on health professionals and reducing their ability to innovate, especially in low- and middle-income countries. It has been established that there is a partial mediation (without the participation of sensory systems) of health between eustress and innovative work behavior, while supervisor support does not mediate eustress and health. In addition, research results show that distress is negatively associated with innovative behavior. At the same time, it is even possible that two opposite emotions (eustress and distress) are manifested in the innovative work behavior of employees. The presence of eustress (“good” stress) exacerbates the harmful effects of a disaster. That is, if we are not sure that distress has been eliminated, eustress can do more harm than good, especially for healthcare professionals who are particularly susceptible to ill health [14,24,25,26].

Against the backdrop of a variety of professional and life stress factors that affect both individuals and teams, resistance to adverse situations has also become an urgent problem in people’s lives. Resilience can be defined as successful and positive adaptation (an individual’s ability to endure, recover or return to the state they were in prior to the situation in question) to varying conditions or in the face of significant and severe levels of stress, trauma and adversity, and the ability to thrive and survive despite hardships [1,7,8]. Specifically, in psychology, resilience is defined in three ways. First, resilience is the positive ability of people to cope with major adversity, trauma, tragedy, threat or major source of stress and disaster. Second, this is the ability to return to homeostasis after a disturbance. Finally, resilience is an adaptive system that uses exposure to stress and disaster to provide resistance to future negative events [8,14].

According to a number of theories of activation (emotional state) that have been formed over the past 100 years, a person goes through a series of cyclic processes every day [27,28,29,30] beginning with sleep phases and progressing to various levels of wakefulness, ultimately reaching a certain level of psycho-emotional stress and even physiological stress [7]. The common feature of these processes is reversibility without any support. Irreversible processes, pathological stress and illness, follow next. These processes do not return to normal on their own. This requires medical or psychological support. Thus, the Human Function Curve was constructed, where a “fatigue point” was set, separating reversible and irreversible processes [8].

The Human Function Curve shows how increased stress arousal—from eustress to distress—affects job performance [8,28,31]. When the level of stress during arousal reaches a certain level (fatigue), performance drops sharply. It is important to note that the fatigue point is different for different people and depends on the psychology of the individual, their awareness as well as on the conditions in which the person is located. As arousal increases (activation of the emotional sphere), no one notices that performance begins to decline and certain symptoms appear, such as exhaustion, health problems (such as headaches and migraines) and, finally, a complete breakdown. Therefore, any definition of stress should include both eustress (“good” stress) and distress (“bad” stress) [29,32,33].

With a very large line of classifications related to stress, factors can be roughly divided into the following six types: (1) crises/disasters, (2) important life events/acute stressors, (3) everyday problems/microstressors (for example, a tire puncture or a small fine), (4) chronic stressors (including ongoing financial stress, marital problems, divorce or academic stress), (5) external stressors (e.g., air conditioning or noise) and (6) organizational stressors (e.g., toxic leadership) [8,15,29]. Of course, these factors can be listed further. More importantly, we see the association of known factors with the emergence of new conditions and technologies [3,5,6].

Taking into consideration the above factors of stress activation, firstly sensory (cognitive and informational) as well as psycho-emotional reactions involved, this research was aimed at the psychological and neurophysiological substantiation of approaches to express algorithms for assessing cognitive functions and resilience to stress under time deficit.

Methodological approaches for assessing the human cognitive status are constantly evolving. At the same time, the creation of new assessment methods is accompanied by traditional research. In particular, modern methods of artificial intelligence make it possible to solve the problem of classifying states and human behavioral activity based on the analysis of external features obtained during the process of analyzing images from a video camera (including a smartphone) and forming conclusions about its internal state [5,6]. However, to prove these findings, I plan to perform interdisciplinary research in the fields of experimental neurophysiology, psychology, data mining, computer vision, ontological modeling, profiling and recommender systems.

## 2. Materials and Methods

For the assessment of cognitive functions, two methodological approaches were applied. The first of them was a complex of psychological testing, and the second consisted of an EEG test with subsequent computer processing of the big data. The full range of examination took up to 40 min with a subsequent reduction to 15–25 min.

Using the selected methodological techniques, a study of three groups of subjects was performed. Study group I consisted of 102 healthy volunteers (18–35 years) who underwent psychological testing. For neurophysiological research, study group II was formed, which consisted of 102 healthy subjects (18–60 years) who underwent a planned psychiatric examination for professional purposes.

To compare the data of healthy subjects, group III was formed, which consisted of 38 outpatients of a neuropsychiatric clinic [31].

### 2.1. Psychological Testing

The complex of psychological testing included six tests: the questionnaire of the severity of psychopathological symptoms (SCL-90-R), the method of clinical and psychological research of the personality structure “Mini-Mult”, as well as a block of logical methods: “Isolation of essential features”, “Exclusion of unnecessary”, “Simple analogies” and “Understanding the figurative meaning of proverbs and metaphors”.

The SCL-90-R test results were used to assess nine main scales of symptomatic disorders: somatization, obsessive-compulsiveness, interpersonal anxiety, hostility, depressiveness, anxiety, phobia, paranoia, and psychoticism [34]. In addition, three second-order SCL-90-R scales were analyzed: Present Symptomatic Distress Index (PSDI), General Severity Index (GSI) and Total Positive Responses (TPR). The function of each of these is to bring the level and depth of personal psychopathology to a single scale. GSI is an indicator of the current state and depth of the disorder. PSDI is solely a measure of the intensity of the condition, corresponding to the number of symptoms. TPR is simply a count of the number of symptoms to which the patient gives positive responses—that is, the number of statements for which the subject marks at least some level above zero [35].

Mini-Mult is an abbreviated version of the Minnesota Multiphasic Personality Inventory (MMPI test, 556 questions) and consists of 71 questions and 11 scales, of which 3 are evaluative [36]. In the present study, we analyzed the rating scales of lie, reliability and correction, which affect the mathematical calculation of thematic scales. In addition, the following basic scales were considered: hypochondria, hysteria, psychopathy, paranoia, psychasthenia, schizoidness and hypomania.

### 2.2. EEG Registration and Processing

EEG at rest with the performance of traditional functional tests in eight bipolar leads: Fp_1_–C_3_, Fp_2_–C_4_, C_3_–O_1_, C_4_–O_2_, O_1_–T_3_, O_2_–T_4_, T_3_–Fp_1_, T_4_–Fp_2,_ according to the international 10/20 system, was recorded. Biosignal processing was carried out using the WinEEG software package by counting EEG indices and power spectra in 5 frequency bands (theta, delta, alpha, beta1 and beta2). The location of the electrodes can be shown by considering the results of the indices and spectra below.

### 2.3. Outcomes Processing

To select the most valid indicators under time constraints, the following algorithm for assessing biosignals was chosen. After dividing the EEG recording interval into segments (analysis epochs), calculations for each channel were performed separately. First of all, for each segment of the EEG recording, the parameters of the polynomial trend were calculated, and this trend was compensated. The order of the polynomial trend was set by the corresponding parameter, and I chose the value equal to 0; that is, only the constant component was eliminated during the calculation. To suppress the leakage of energy through the side maxima, each segment was smoothed out via a time window. Bartlett, Khann and Welch time windows could be used for this. During my work, I chose to use the Hann time window. Further, using the “fast Fourier transform”, the power spectrum (periodogram) was calculated.

The results of the study using the statistical program SPSS for Windows according to the following algorithm were analyzed. Initially, each sample was checked for compliance with the normal distribution (Gaussian) visually using the construction of a histogram and using the Kolmogorov–Smirnov criterion. The data obeying the laws of normal distribution were analyzed for reliability using Student’s *t*-test. Data for which compliance with the normal distribution could not be proved were analyzed using the Wald–Wolfowitz and Mann–Whitney tests. As a result, indicators were selected that were statistically different in the studied groups [4,31].

As part of analytical statistics, a multivariate analysis was performed to assess the dependence of one quantitative trait on several other traits when predicting the value of one trait based on the value of several traits. Since the dependent and independent features were quantitative, I chose the method of multiple linear regression analysis, in which each of the studied features was consistently evaluated.

## 3. Results

### 3.1. Psychological Testing

When analyzing the results of clinical and psychological testing from SCL-90-R, scales and indicators were revealed that differed to a statistically significant degree in subjects in study groups I (conditionally healthy) and III (outpatients). These are the second-order scales of SCL-90-R—the general index of severity (mean value in group I: 0.73 ± 0.04; in group III: 1.08 ± 0.12) and the total number of affirmative answers (mean value in group I: 1.46 ± 0.03; in group III: 1.9 ± 0.17).

Statistically reliable results for most scales of the Mini-Mult method were obtained (Table 1).

High results (the level of statistical significance *p* < 0.01 according to the Mann–Whitney and Wald–Wolfowitz criteria) were obtained when analyzing the test data using the methods “Excluding unnecessary”, “Simple analogies”, “Isolation of essential features” and “Understanding the figurative meaning of proverbs and metaphors”.

Test results according to the “Excluding unnecessary” method in study group III consisted of 5 points in 4 people and 3 points or less in 5 people (average value 4.97 ± 0.02), while test results in study group I had the prevailing mark of 5 points (mean value 3.8 ± 0.02).

According to the “Simple analogies” method in study group III, the highest score of 23 points is absent and 11 patients scored less than 20 points (mean value 20.7 ± 1.8), while in study group I, 23 points prevail (mean value 22.7 ± 0.06).

Testing “Isolation of essential features” showed the following results. In study group III: more than 15 points for 4 people and 13 points or less for 7 people (mean value 13.55 ± 0.3). In study group I, scores of 15–16 points prevail (mean value 15.87 ± 0.04).

According to the methodology “Understanding the figurative meaning of proverbs and metaphors”, the following data were obtained: in study group III, 9 patients scored 7 points or less (mean value 7.55 ± 0.3), while in study group I, the score was 9 points (average value 8.99 ± 0.01).

For the results of psychological testing, a multivariate regression analysis was carried out, which made it possible to clarify the picture of the relationship between the studied methods and scales. It was found that the integrative indicator “general index of severity” of the SCL-90-R test shows a stable correlation with all thematic scales of this technique. Figure 1 shows a graph of standard regression coefficients (ß) in relation to the dependent feature “general index of severity”.

It can be assumed that the integrating role of this indicator in the aggregate analysis of all thematic scales is quite significant. The combination of this conclusion with the statistical reliability of the indicator proved above makes it sufficiently representative for judging within the framework of “norm” or “pathology” in conditions of lack of time. A similar correlation between the symptom severity indexes has been described in a line of other studies.

Figure 2 displays a scatterplot with regression lines showing a linear relationship between Depression Scale scores and the GSI score—General Severity Index (standard regression coefficient is 0.907).

It has been shown that many scales of the Mini-Mult method have a regression dependence on the second-order scale “Index of present symptomatic distress”. SCL-90-R, for example, shows this regression along the scale of psychopathy (see Figure 2). The correlation between different scales has been proved not only within the framework of the same methods, but also between individual indicators of all the tests I used. In particular, the results of the scale of hypochondria Mini-Mult correlate with the indicators on the scale of somatization SCL-90-R. The results of the scale of psychasthenia Mini-Mult correlate with indicators on the scale of interpersonal anxiety SCL-90-R.

When conducting a multiple linear regression analysis of test results using the methods “Exclusion of the superfluous”, “Simple analogies”, “Isolation of essential features” and “Interpretation of the figurative meaning of proverbs and metaphors”, it was found that for three tests at once, the dependent factor with the maximum standard regression coefficient is technique “Interpretation of the figurative meaning of proverbs and metaphors.” The effectiveness of this test has also been demonstrated in a number of publications; it has been reported that subjects with diseases of the central nervous system often cannot understand the figurative meaning of proverbs and metaphors.

A scatter diagram with regression lines showing a linear regression relationship for the test results using the “Simple analogies” and “Interpretation of the figurative meaning of proverbs and metaphors” (standard regression coefficient is 0.355) is shown in Figure 3.

### 3.2. EEG Registration and Processing

In a neurophysiological study, resting EEG was recorded for a minute in eight bipolar leads: Fp_1_–C_3_, Fp_2_–C_4_, C_3_–O_1_, C_4_–O_2_, O_1_–T_3_, O_2_–T_4_, T_3_–Fp_1_, T_4_–Fp_2_. Subsequent computer processing of the signal was carried out using the WinEEG software package by calculating the EEG power indices and spectra in five frequency ranges (theta, delta, alpha, beta1 and beta2).

Attention is drawn to the significant difference in the average results in the theta range; for example, for the C_4_–O_2_ derivation, the average value in study group II is 6.31 ± 0.4, and in study group III: 17.63 ± 2.9 for the delta range in that group. In the same assignment, the mean value in study group II was 6.60 ± 0.5, and in study group III, it was 11.95 ± 2.1 (Table 2).

Table 3 shows the average values in study groups II and III obtained by calculating the EEG power spectra in the alpha and beta1 ranges. For the C_4_–O_2_ site in the alpha range, the average value in study group II was 69.43 ± 1.7, and in study group III, it was 47.84 ± 4.9. In the beta1 range, the average value in study group II was 6.15 ± 0.6, and in study group III, it was 8.48 ± 1.4. For site O_1_–T_3_ in the alpha range, the average value in study group II was 66.15 ± 1.9, and in study group III, it was 46.30 ± 4.4. In the beta1 range, the average value in study group II was 6.32 ± 0.6, and in study group III, it was 8.91 ± 1.5.

EEG indices with significant differences in the studied groups are the temporo-frontal areas in the delta range. For EEG power spectra, a significant statistical difference in the results of the compared groups was shown for the occipital right hemispheric area in the delta range, for the occipital sites and in the left and right hemispheres for the alpha range, as well as for the temporal–frontal sites for the beta1 range. Table 4 and Table 5 show the statistical significance of the results of comparing the indices and spectra of the main EEG rhythms according to the Wald–Wolfowitz and Mann–Whitney criteria. In cases of *p* > 0.05, the significance level is not given.

Figure 4 (left) shows a line chart for the values of the EEG indices for the theta range in lead Fp_1_–C_3_ (the red curve shows the distribution of the trait in healthy subjects of group II and the blue curve shows the distribution of the train in outpatients of group III). Figure 4 (right) shows a linear diagram for the values of the EEG power spectra for the theta range in lead C_3_–O_1_ (the red curve shows the distribution of the sign in healthy subjects of group II and the blue curve shows the distribution of the sign in outpatients of group III).

Thanks to linear regression analysis, the relationship between indices and power spectra was studied in detail. The data of the linear regression analysis of indicators in all studied ranges indicate a stable dependence of signs within a specific range: for example, in the form of a correlation between changes in “paired sites” (I call two bipolar sites “paired” if points are a mirror image of each other at different hemispheres). Interhemispheric interactions and correlations between frequency scores between the left and right hemispheres were also described previously.

For a number of EEG indices and power spectra in the delta range, a regression relationship was established with the same-name indicators in the alpha range. The projection of the obtained regression equations onto the scalp is curious, since the delta-alpha conjugation of the indices is characteristic of the posterior hemispheric leads, and the delta-alpha conjugation of the power spectra is characteristic of the anterior hemispheric leads (see Figure 5). Moreover, in this case, the obtained regression coefficients are negative.

In order to reduce the dimensions of the studied multidimensional trait and select the most informative indicators, a factor analysis was performed. Of all the factors selected by the statistical program, based on the scatter diagram (see Figure 6), only those whose values exceeded one were selected. Such a diagram serves to separate unimportant factors from the most significant factors. These significant factors form a “slope” on the graph—the part of the line characterized by a steep rise. In the present chart, there is a steep rise in the region of the first five factors. These five factors formed the basis of the model at the beginning of the factor analysis.

At the next stage of factor analysis, factor loadings for each of the factors I have chosen were considered. As a result, the following definitions of factors were obtained: factor 1—indices and spectra in Delta and Alpha ranges; factor 2—indices in the Beta1 and Beta2 ranges; factor 3—power spectra in the Beta1 range; factor 4—power spectra in the Beta2 range; factor 5—indices and spectra in the Theta range in the frontal and temporal sites.

A statistically significant difference with an error level of *p* < 0.01 according to the Wald–Wolfowitz and Mann–Whitney tests was proven for factors 1 and 5.

Figure 7 shows a scatterplot with a direct regression illustrating the relationship between the EEG indices in site O_1_–T_3_ in the delta and alpha ranges.

## 4. Discussion

While the tests used in this study were originally developed for clinical research, there is no doubt that a perfectly healthy person is also exposed to stress factors with a variety of manifestations. However, this may not always be a sign of a mental disorder. More often, a person (or a professional team) develops resilience to stress [8,31,34,35].

SCL-90-R allows for the evaluation of psychopathological symptoms, or, in other words, reflects the stress reactivity of the individual tested: the specifics of his reaction to the stress factors of the surrounding world. Accordingly, the idea of a “norm” in this case also changes in relation to the definition of “mental health”. At the same time, general indicators integrating unfavorable combinations of high values obtained, such as the overall symptom severity index and the symptomatic distress index, were more sensitive when comparing subjects from different study groups. A line of authors also describes the high validity of the integrative indicators of the SCL-90-R [36,37,38,39,40].

Analysis of the results of clinical and psychological testing of the SCL-90-R established scales and indicators that were different to a statistically significant degree for the subjects in study groups I (healthy) and III (outpatients). These are the second-order scales SCL-90-R—the overall symptom severity index (mean value in group I: 0.73 ± 0.04 and in group III: 1.08 ± 0.12) and the index of present symptomatic distress (mean value in group I: 1.46 ± 0.03 and in group III: 1.9 ± 0.17).

According to researchers, the most informative indicator is the second-order index of the presence of symptomatic distress indicator, which was used to quickly assess the subjects [36,37,38]. Some studies indicate the statistical significance of all integrative indicators and indicators on the anxiety scale [38,41]. In other studies, the authors emphasize the significance of test results on all scales. For example, students who spend a significant amount of time on the Internet, using SCL-90-R, showed significant differences compared to the control group on all thematic scales [35,42]. SCL-90-R also proved to be effective in examining patients with various somatic diseases, such as in patients suffering from pelvic pain syndrome [35].

Thus, for SCL-90-R in the present study, the second-order integrative indicators are of the greatest importance, including the general index of symptom severity (GSI) and the index of present symptomatic distress (PSDI). At the same time, analysis of the results of individual scales showed that only one of the nine scales (depression) was independently statistically significant. Similar outcomes have been reported using the SCL-90-R by a line of authors, for example, to assess stress reactivity in college students. According to researchers, the second-order indicator GSI (general symptom severity index) is the most informative, which can be used to quickly assess subjects [36].

The effectiveness of using the Mini-Mult method in conditions of time shortage is determined by the results on the scales of hypochondria, depression, hysteria, paranoia, psychasthenia, schizoid and hypomania, where the differences in the studied groups have a reliability of *p* < 0.05. High values (>70) in any of the Mini-Mult scales require more in-depth examination. Such a survey, which takes 15–25 min, can be used as a standard for assessing neuro-cognitive status in screening studies of large collectives.

Only for one thematic scale—the psychasthenia scale—was it not possible to prove a statistically significant difference. In the published research, the attitude towards the reliability of the use of this test is rather ambiguous. Some authors insist on the insufficient validity of the reduced version of the Mini-Mult personality structure study compared to the “full version” of the MMPI when applied to psychiatric patients [40], or when examining parents with problems in raising children [43]. At the same time, a number of other studies show good results using the Mini-Mult; the effectiveness of Mini-Mult testing in psychiatric patients is noted precisely in conditions of time pressure, which was of fundamental importance in this work [44]. Like the SCL-90-R, the Mini-Mult test is also successfully used in clinical medical practice, such as when assessing the mental status of patients with the consequences of acute myocardial infarction [45].

In our opinion, Mini-Mult allows one to study the structure of the personality, and, apparently, operates on deeper and more permanent components of the cognitive status, unlike SCL-90-R, which actualizes sensations and perceptions now, at the moment, in relation to research. A healthy person can be under stress, and visiting a doctor does not always give him positive emotions. However, the question of psychopathology is not relevant if a person is able to switch from negative to positive components and be distracted; that is, if they are able to have a certain mobility of mental processes. On the other hand, when mental processes form stable dominants that exist not only today and now but also long before the study due to the duration of their manifestations, this can lead to personality traits that are better diagnosed using Mini-Mult. The effectiveness of the Mini-Mult test for assessing mental status is also confirmed by literature data [45].

High results (the level of statistical error *p* < 0.01 according to the Mann–Whitney and Wald–Wolfowitz criteria) were obtained in the analysis of the logical tests “exclusion of the superfluous”, “simple analogies”, “essential features” and “interpretation of the figurative meaning of proverbs and sayings “. Psychological tests that assess logical thinking in different variations are widely described in the literature, and their effectiveness in complex studies is especially emphasized [46,47]. When conducting a multiple linear regression analysis of the test results using the “exclusion of unnecessary”, “simple analogies”, “essential features” and “figurative meaning” methods, it was found that for three methods at once, the dependent factor with the maximum standard regression coefficient is the method “interpretation of the figurative meaning of proverbs and sayings”. The effectiveness of this test has also been demonstrated in a line of publications; it is described that subjects with diseases of the central nervous system often cannot understand the figurative meaning of proverbs and metaphors [31].

The outcomes of this research indicate that the indices and power spectra at the same registration points for the same EEG range, as a rule, are related. In percentage terms (so that the sum of 100% is obtained by adding up the indicators in all ranges), the indices and power spectra reflect the “degree of presence” in the total oscillations of waves of a particular range. Most of the recorded EEG parameters are characterized by a relationship between the indicators in the “adjacent leads”, since, in a bipolar study, the recorded curve reflects the algebraic sum of fluctuations in the electric potential under two electrodes. Thus, one of the two components of the algebraic sum for each of the pairs of “adjacent leads” turns out to be common. In addition, in some cases (this is shown for the delta, beta-1 and beta-2 ranges), there is a regression relationship between “paired electrodes” for identical leads in different hemispheres. We can assume that the delta and beta components of the oscillations are symmetric in the left and right hemispheres.

These approaches may be used to develop new efficient systems for the automatic classification of EEG [3,4,16]. For example, developing a technology to distinguish normal from pathological EEG as well as to classify different types of pathology and different types of human functional states triggered by stressors and sensory stimulations is possible. Data from a 30 s EEG assessment paired with a small amount of other data can be sent easily to remote computers for quick analyses and results. This may be critically important when the timely receipt of such results is important [2,31].

The “psychological situation” perceived by the person is considered critical for determining the stress level. Based on this, “cognitive assessment”, including primary and secondary assessment, is important for a person under time pressures (lack of time) [26]. During the initial assessment, a judgment is formed about the significance of the event as stressful, positive, controlled, provocative or inadequate, followed by an assessment of one’s resources and survival opportunities. The secondary assessment is the activation of memory processes about a potential reaction to a threat and its being overcome as a process of fulfilling this reaction [12,13].

As a rule, in publications related to the study of the mechanisms of regulation of sensory signals of varying intensity entering the brain, a connection is made with the processes of stimulation/inhibition in the central nervous system [27,28]. It is assumed that the nature of the response depends on the threshold sensitivity. The presence of low sensitivity, i.e., an initially high threshold of absolute sensory sensitivity, increases the severity of the reaction to an increase in the intensity of the stimulus. On the contrary, a system with high sensitivity, that is, with a low absolute sensory threshold, launches a “program” that protects against “overload”, and, despite an increase in stimulus intensity, a reduced evoked response is obtained [2,28].

Reactions to stimuli in general and stressful challenges in particular for healthy people are subject to a number of well-known laws and regulations described by psychologists and physiologists over the past 100 years [27,28,31]. The impact of dynamic factors of the external and internal environments contributes to the formation of adaptive rearrangements in the central and peripheral nervous system, as well as in the endocrine and sensory systems. This is reflected in the implementation of attention, memory, consciousness and other cognitive functions. The universal mechanism explaining dynamic shifts is the law of force and, as a stage of its development, limiting protective inhibition. The mechanism of protective inhibition, which makes it possible to explain compensatory changes in pathology, leaves a number of questions in the provision of mental functions under extreme conditions. When studying reflex activity, the hypothesis of “preventive” inhibition, which occurs under the action of weak stimuli on a healthy organism, was proposed as a solution to the question posed [26,31]. In sensory physiology, this question received the most complete analysis when studying the phenomenon of augmentation/reduction (A/R) [28].

One of the best-known neurophysiological theories of personality is the Reinforcement Sensitivity Theory (RST). The original formulation of the RST emphasized only two neurobehavioral systems: the behavioral inhibition system (BIS) and the behavioral approach system (BAS). There is an underlying complexity between these systems, and this is reflected in the Revised RST (rRST), which postulates three main neuropsychological systems: two of which are responsible for defensive behavior (the fight-flight-freeze system, FFFS and the BIS) and is responsible for approach behavior (BAS). In its original form, RST is associated with a kinesthetic psychological phenomenon called increase/decrease (A/R). In such situations, some persons are more inclined to increase the virtual weight of the object, while others are more inclined to decrease it.

Undoubtedly, the considered direction of research on the search for markers of stress resistance can and will continue to develop indefinitely. This is determined by methodological techniques, stress factors, working conditions in professional teams, etc. Based on the existing concepts of the development of stress reactions for express analysis (25-40 min) of stress resistance, it will be mandatory to assess the sensory (cognitive and informational) components of stress, as well as the study of psycho-emotional reactions. As a variant, the present paper substantiates the research algorithm consisting of clinical and psychological testing, as well as EEG with traditional tests and analysis of indicators and spectra.

### Limitations

Patients from group III did not receive active pharmacological treatment a week before the planned study.

All participants of the research project signed a voluntary informed consent.

## 5. Conclusions

Integrative index SCL-90-R is the general index of severity of symptoms and has a high statistical significance (*p* < 0.05). In the group of healthy persons, the average value of the indicator was 0.73 ± 0.04, whereas in the group of neuropsychiatric outpatients, the average value was 1.08 ± 0.12;The effectiveness of using the Mini-Mult method in conditions of time shortage is determined by the results on the scales of hypochondria, depression, hysteria, paranoia, psychasthenia, schizoid and hypomania, where the differences in the studied groups have a reliability of *p* < 0.05. High values (>70) in any of the Mini-Mult scales require more in-depth examination. Such a survey, which takes 15–25 min, can be used as a standard for assessing neuro-cognitive status in screening studies of large collectives;Logical methods, including testing “understanding the figurative meaning of proverbs and metaphors” are highly informative in assessing the mental status in conditions of time deficit. High statistical significance was proved for all methods (*p* < 0.01). In the test “Understanding the figurative meaning of proverbs and metaphors”, the average value in the group of healthy subjects was 8.99 ± 0.01, while in the group of neuropsychiatric outpatients, it was 7.55 ± 0.3;Indices and spectra of EEG power in the theta, delta and alpha frequency ranges are an effective reflection of cognitive status. EEG power indices and spectra consistently correlate with each other in bipolar derivations in all frequency ranges, with regression coefficients for the correlation dependence in the range of 0.6–0.8. Data on indices and power spectra in the theta range are highly reliable (*p* < 0.01) in assessing neurophysiological status.

## Figures and Tables

**Figure 1 behavsci-13-00258-f001:**
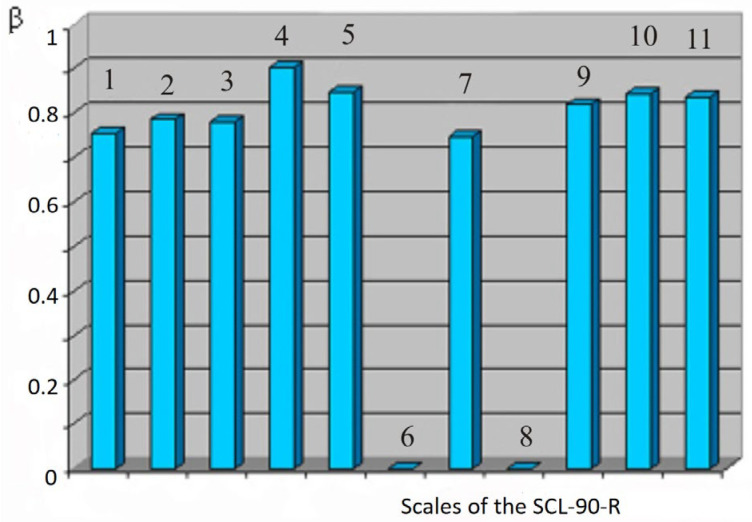
Standard regression coefficients for the SCL-90-R “General Severity Index Scale”. Notes: The ordinate is the standard regression coefficients (ß). On the abscissa axis of the scale 1—somatization, 2—obsessiveness-compulsiveness, 3—interpersonal anxiety, 4—depression, 5—anxiety, 6—hostility, 7—phobias, 8—paranoia, 9—psychoticism, 10—the total number of affirmative answers, 11—index of present symptomatic distress.

**Figure 2 behavsci-13-00258-f002:**
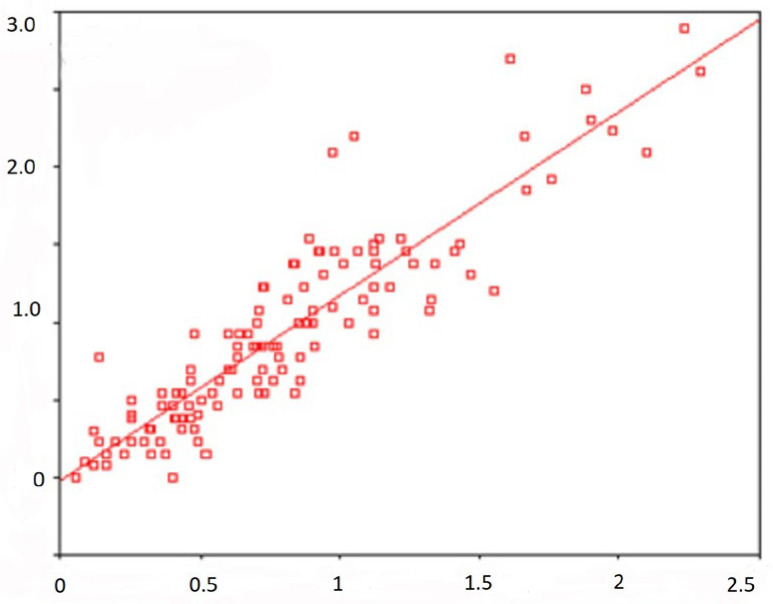
Scatter diagram. Note: Along the ordinate axis is the depression scale (in points); along the abscissa axis is the general severity index (in points).

**Figure 3 behavsci-13-00258-f003:**
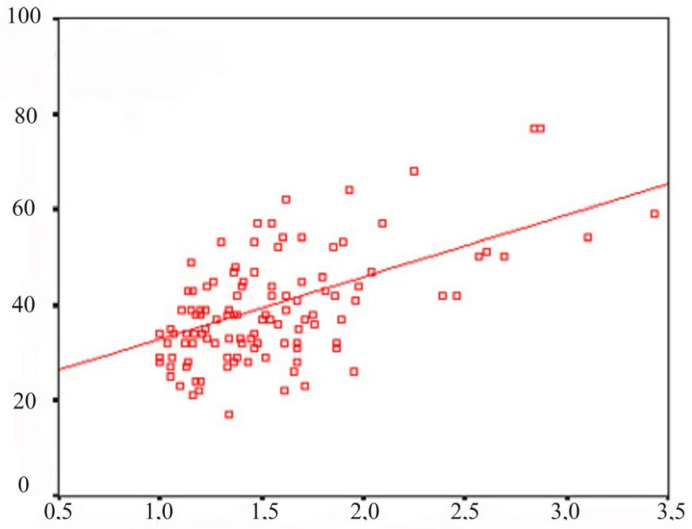
Scatter plot. The ordinate is the psychopathy scale (points) and the abscissa is the SCL-90-R symptomatic distress index (points).

**Figure 4 behavsci-13-00258-f004:**
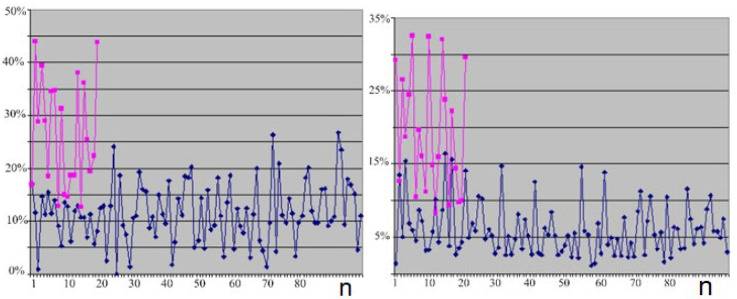
EEG index line diagram (%) in theta band (4–7 Hz) for sites Fp_1_–C_3_ (**left**) and EEG power spectrum line diagram (%) in theta range for sites C_3_–O_1_ (**right**). Note: n—Number of studies; red line—healthy persons, blue line—outpatients.

**Figure 5 behavsci-13-00258-f005:**
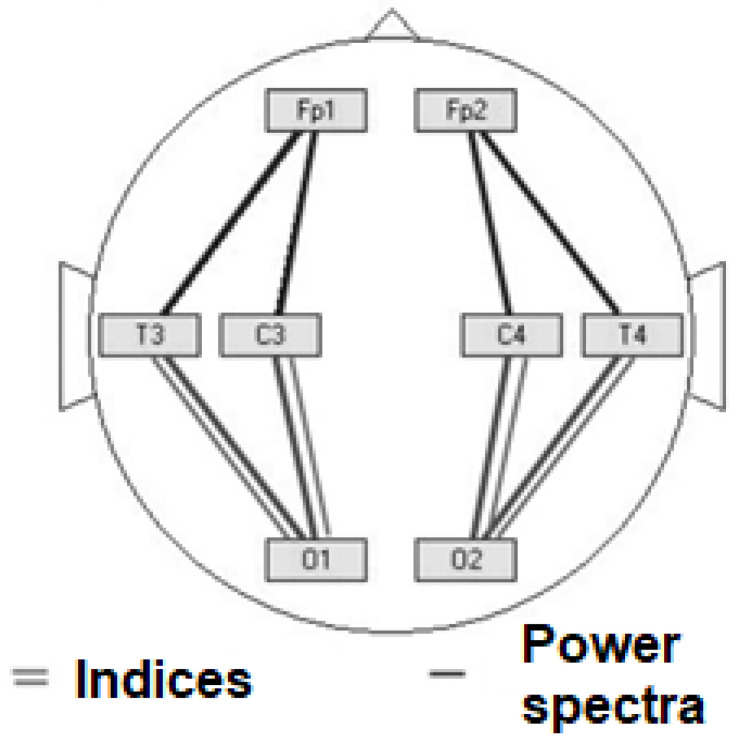
Localization of the identified relationships in the delta and alpha ranges.

**Figure 6 behavsci-13-00258-f006:**
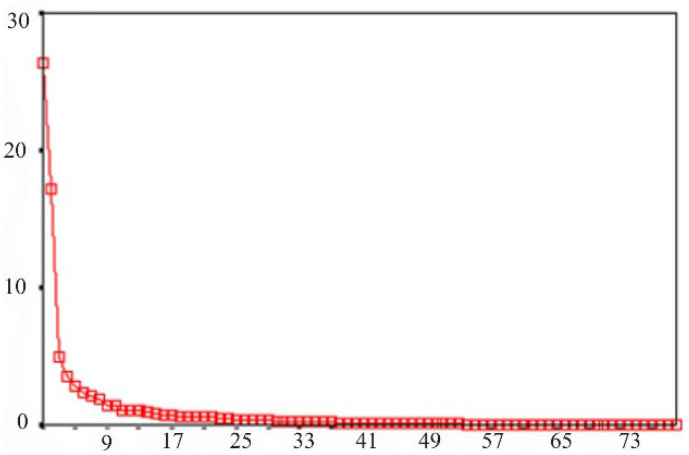
Scatter plot of factor analysis of the spectral power of the main EEG rhythms. Note: on the ordinate—the value of the factor; on the abscissa—the number of the factor. Note: explanations in the text.

**Figure 7 behavsci-13-00258-f007:**
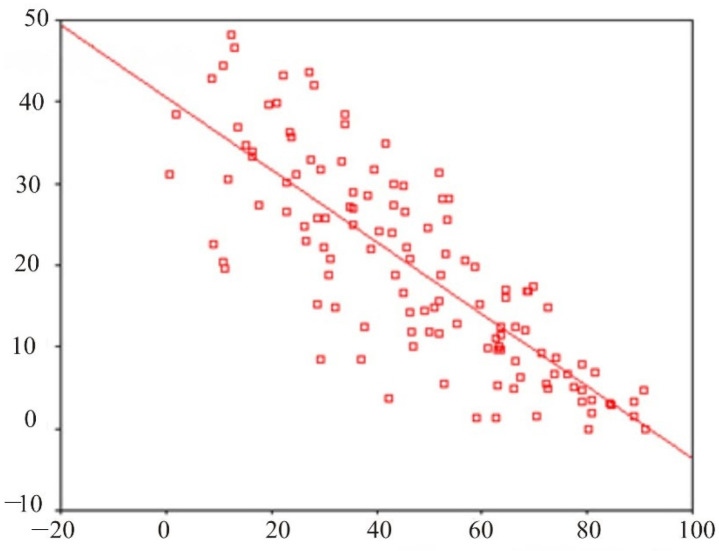
Scatter diagram of the direct regression illustrating the relationship between the EEG indices in site O_1_–T_3_ in the delta and alpha ranges.

**Table 1 behavsci-13-00258-t001:** Test results according to the Mini-Mult method, as points.

Scales	Mean ± Error of the Mean	
	Study Group I	Study Group III
Lie	41 ± 0.9	45.9 ± 2.4
Hypochondria	55 ± 0.87	58.1 ± 2.3
Depression	46 ± 1.1	65.9 ± 3.0
Hysteria	41 ± 1.0	61.3 ± 2.3
Paranoia	34 ± 1.35	52.0 ± 3.1
Psychasthenia	55 ± 0.90	59.9 ± 2.7
Schizoid	52 ± 1.00	60.9 ± 3.3
Hypomania	43 ± 1.06	51.7 ± 2.7

**Table 2 behavsci-13-00258-t002:** EEG power spectra (%) in delta and theta ranges.

Sites, 10/20	Delta		Theta	
	Group II	Group III	Group II	Group III
Fp_1_–C_3_	14.72 ± 0.9	13.45 ± 2.4	10.38 ± 0.5	16.32 ± 2.2
Fp_2_–C_4_	13.18 ± 0.8	13.80 ± 2.8	10.28 ± 0.6	16.93 ± 2.4
C_3_–O_1_	7.30 ± 0.6	11.40 ± 2.0	6.16 ± 0.4	17.49 ± 2.9
C_4_–O_2_	6.60 ± 0.5	11.95 ± 2.1	6.31 ± 0.4	17.63 ± 2.9
O_1_–T_3_	9.60 ± 1.3	12.60 ± 2.3	6.02 ± 0.4	16.74 ± 2.5
O_2_–T_4_	7.52 ± 0.5	12.83 ± 1.7	6.12 ± 0.4	17.79 ± 2.8
T_3_–Fp_1_	14.25 ± 0.9	12.70 ± 2.2	10.46 ± 0.5	18.63 ± 2.7
T_4_–Fp_2_	13.66 ± 0.8	13.22 ± 1.7	11.17 ± 0.6	18.72 ± 2.3

Note. The mean ± error of the mean.

**Table 3 behavsci-13-00258-t003:** EEG power spectra (%) in alpha and beta1 bands.

Sites, 10/20	Alpha		Beta1	
	Group II	Group III	Group II	Group III
Fp_1_–C_3_	47.83 ± 2.0	40.4 ± 4.4	5.79 ± 0.4	9.40 ± 1.5
Fp_2_–C_4_	50.05 ± 2.0	42.05 ± 4.7	5.73 ± 0.4	9.85 ± 1.6
C_3_–O_1_	67.39 ± 1.9	47.98 ± 4.6	5.98 ± 0.6	8.83 ± 1.5
C_4_–O_2_	69.43 ± 1.7	47.84 ± 4.9	6.15 ± 0.6	8.48 ± 1.4
O_1_–T_3_	66.15 ± 1.9	46.30 ± 4.4	6.32 ± 0.6	8.91 ± 1.5
O_2_–T_4_	68.26 ± 1.7	45.94 ± 5.0	6.24 ± 0.6	8.09 ± 1.3
T_3_–Fp_1_	47.70 ± 1.7	37.49 ± 4.1	6.86 ± 0.5	9.90 ± 1.5
T_4_–Fp_2_	47.90 ± 1.7	39.70 ± 3.8	6.56 ± 0.5	10.80 ± 1.7

Note. The mean ± error of the mean.

**Table 4 behavsci-13-00258-t004:** Statistical significance of EEG indices.

Sites, 10/20	Frequency Bands				
	Delta	Theta	Alfa	Beta1	Beta2
Fp_1_–C_3_	*	*p* < 0.01	*	*	*
Fp_2_–C_4_	*p* < 0.05	*p* < 0.01	*	*	*
C_3_–O_1_	*	*p* < 0.01	*	*	*
C_4_–O_2_	*	*p* < 0.01	*	*	*p* < 0.05
O_1_–T_3_	*	*p* < 0.01	*	*	*p* < 0.05
O_2_–T_4_	*	*p* < 0.01	*p* < 0.05	*	*
T_3_–Fp_1_	*p* < 0.05	*p* < 0.01	*	*	*p* < 0.05
T_4_–Fp_2_	*p* < 0.05	*p* < 0.01	*	*	*

Note. *—differences are not significant (*p* > 0.05).

**Table 5 behavsci-13-00258-t005:** Statistical significance of EEG power spectra.

Sites, 10/20	Frequency Bands				
	Delta	Theta	Alfa	Beta1	Beta2
Fp_1_–C_3_	*	*p* < 0.01	*	*p* < 0.05	*
Fp_2_–C_4_	*	*p* < 0.01	*	*	*
C_3_–O_1_	*	*p* < 0.01	*p* < 0.05	*p* < 0.05	*
C_4_–O_2_	*p* < 0.05	*p* < 0.01	*p* < 0.05	*	*
O_1_–T_3_	*	*p* < 0.01	*p* < 0.05	*	*
O_2_–T_4_	*p* < 0.05	*p* < 0.01	*p* < 0.05	*	*
T_3_–Fp_1_	*	*p* < 0.01	*	*p* < 0.05	*
T_4_–Fp_2_	*	*p* < 0.01	*	*p* < 0.05	*

Note.*—differences are not significant (*p* > 0.05).

## Data Availability

Data available on request due to restrictions of the privacy and ethical.

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
