# Peer review of "Psychological and Neurophysiological Screening Investigation of the Collective and Personal Stress Resilienceâ€"

_behavsci, 2023, doi:10.3390/bs13030258_

Round 1
Reviewer 1 Report
Journal: “Behavioral sciences”
Type of Paper: Article
Title of the manuscript: “Psychological and Neurophysiological Screening Investigation of the Collective and Personal Stress Resilience ”.
This study reports the results of an EEG study conducted to evaluate the neurophysiological patterns of stress resilience. The topic proposed in the title is very interesting and topical. However, despite the catchy title, I found the manuscript confusing and at times not comprehensible.
In the following, I will try to provide some specific suggestions per section to improve it.
ABSTRACT: the abstract is confusing and one struggles to understand the rationale behind the study, the research objective, methods and results. I recommend rewording it to better follow the journal guidelines.
INTRODUCTION: overall, the introduction is very confusing and it is difficult to follow the author's logical thread. Some periods are not understandable and there is a lack of references for the statements made.
Lines 30-34: rephrase period and insert reference.
Lines 34-35: reference to the statement "However, the using..... are quite expensive" is missing.
Line 37: the subject of the study, i.e. stress, is introduced: the author should provide a clear definition and reference.
Lines 46-47: sentence is unclear and lacks references (e.g.: We percive stress factors= what is meant? Does the author refer to the subject's perception or how stress is analysed?).
Lines 54-59: Period to be rewritten and provide supporting literature for the statements.
Lines 65-72: Insert references, if the reference bibliography is only [6, 14] I recommend expanding it and specifying it better in the text. For example: "...factors can be roughly divided into the following six types" [source of classification].
Lines 70-72: I am sorry but the statements seem superficial to me if not supported by recent bibliography on the subject, so they appear only as general opinions of the writer.
Lines 74-83: period in need of reworking with precise definitions and supporting references. For example: "...research shows that AI also .... responsibility": which research? Cite source
Line 79: "Based on the theory of rooting...". : it is taken for granted that the reader knows the theory. It should be briefly introduced and contextualised.
Line 82: sentence not understandable
Lines 84-90: If the intent of the period was to introduce the cognitive and diagnostic aspects related to stress, I recommend resetting it because it is very confusing, and one perceives some confusion between cognitive diagnostics, executive functions, and cognitive skills.
Line 91-99: this period is unrelated to the previous one. Moreover, no definition of the human factor curve is given, and no references are included.
Line 94: What is meant by "fatigue point"?
Line 96: what is meant by arousal? Allow me to observe that the author inserts specific terms taking it for granted that the reader knows the meaning.
Line 100 to the end of the introduction (line numbering is blocked): the period should be better contextualised by inserting the necessary references. Especially since it leads to the description of the aim of the study. The latter "... this research was aimed at the psychological and neuropsychological substation of approaches to express algorithms for assessing cognitive functions and resilience to stress under time deficits" is unclear and needs complete revision. Furthermore, concepts such as "resilience" are included. and "time deficit" that has not been introduced before.
MATERIALS AND METHODS: This section should be rewritten by reorganising it so that the experimental sample, the characteristics of the sample and the methodology used for the cognitive and neurophysiological data are precise.
The characteristics of the group are partially included at the end of the paragraph ("Study group consisted of ....).
In addition, the sources of the instruments used (e.g. SCL-90 R, Mini Mult) should be included for validity.
The procedure for the acquisition of electroencephalographic data is unclear, and the motivation for the choice of specific indicators is not explained '....to select the most valid indicators under time constraints... ).
Statistical analysis: the methodology applied needs to be clarified. Furthermore, an algorithm is referred to but needs to be described.
RESULTS: should be reported more clearly with reference to the methodology described in the previous paragraph.
DISCUSSION: the discussion is unclear in places, and the periods are unconnected; they should be rewritten with more contextualisation. Understand the results point by point and discuss them concerning the figures created.
Reviewer 2 Report
Comments to the Authors
1. The novelty and clear hypothesis of the study can be mentioned at the end of the Introduction section.
2. The Exclusion and Inclusion criteria of the study can be mentioned.
3. In the Material and Method section, some of the methods can be given a heading.
4. Conclusion can be briefed, highlighting the major outcomes of the results, if possible.
5. Some of the older references can be updated with recent ones.
6. Limitations and future perspective of the study can be mentioned.
Round 2
Reviewer 1 Report
Dear author,
as is customary in responding to reviewers' comments, I would ask you to send a point-by-point response with respect to my comments so that I can better verify the changes made to the manuscript. By submitting the manuscript in the revision mode as done , the follow-up work is very time consuming and I may not be able to understand whether the manuscript has been improved or not.
Thank you
Author Response
Dear reviewer!
We admire your detailed work. This helped with the fix. Below I present the work done 2 days ago, point by point – red fonts in the answer.
If the system asks for a paper option, I will upload the Feb 26 paper.
Sincerely,
Author
Journal: “Behavioral sciences”
Type of Paper: Article
Title of the manuscript: “Psychological and Neurophysiological Screening Investigation of the Collective and Personal Stress Resilience ”.
This study reports the results of an EEG study conducted to evaluate the neurophysiological patterns of stress resilience. The topic proposed in the title is very interesting and topical. However, despite the catchy title, I found the manuscript confusing and at times not comprehensible.
In the following, I will try to provide some specific suggestions per section to improve it.
ABSTRACT: the abstract is confusing and one struggles to understand the rationale behind the study, the research objective, methods and results. I recommend rewording it to better follow the journal guidelines.
Abstract changed
INTRODUCTION: overall, the introduction is very confusing and it is difficult to follow the author's logical thread. Some periods are not understandable and there is a lack of references for the statements made.
The introduction as a whole has been modified in accordance with your points. Details below.
Lines 30-34: rephrase period and insert reference.
Phrases have been changed, new links inserted.
Lines 34-35: reference to the statement "However, the using..... are quite expensive" is missing.
Phrase changed, new links inserted.
Line 37: the subject of the study, i.e. stress, is introduced: the author should provide a clear definition and reference.
The paragraph is supplemented with new phrases with links.
Lines 46-47: sentence is unclear and lacks references (e.g.: We percive stress factors= what is meant? Does the author refer to the subject's perception or how stress is analysed?).
The paragraph is supplemented with new phrases with links.
Lines 54-59: Period to be rewritten and provide supporting literature for the statements.
Paragraph modified with new references.
Lines 65-72: Insert references, if the reference bibliography is only [6, 14] I recommend expanding it and specifying it better in the text. For example: "...factors can be roughly divided into the following six types" [source of classification].
Paragraph changed, moved below with new references.
Lines 70-72: I am sorry but the statements seem superficial to me if not supported by recent bibliography on the subject, so they appear only as general opinions of the writer.
Paragraph changed, moved below with new references.
Lines 74-83: period in need of reworking with precise definitions and supporting references. For example: "...research shows that AI also .... responsibility": which research? Cite source
The paragraph has been changed, moved below with the addition of new references on the theory of rooting. We see this theory as an option for the analysis of multimodal data.
Line 79: "Based on the theory of rooting...". : it is taken for granted that the reader knows the theory. It should be briefly introduced and contextualised.
See answer for lines 74-83.
Line 82: sentence not understandable
See answer for lines 74-83.
Lines 84-90: If the intent of the period was to introduce the cognitive and diagnostic aspects related to stress, I recommend resetting it because it is very confusing, and one perceives some confusion between cognitive diagnostics, executive functions, and cognitive skills.
The paragraph has been removed.
Line 91-99: this period is unrelated to the previous one. Moreover, no definition of the human factor curve is given, and no references are included.
The paragraph has been changed, moved lower, supplemented with information and references to activation (emotion) theories.
Line 94: What is meant by "fatigue point"?
See answer for lines 91-94.
Line 96: what is meant by arousal? Allow me to observe that the author inserts specific terms taking it for granted that the reader knows the meaning.
See answer for lines 91-94.
Line 100 to the end of the introduction (line numbering is blocked): the period should be better contextualised by inserting the necessary references. Especially since it leads to the description of the aim of the study. The latter "... this research was aimed at the psychological and neuropsychological substation of approaches to express algorithms for assessing cognitive functions and resilience to stress under time deficits" is unclear and needs complete revision. Furthermore, concepts such as "resilience" are included. and "time deficit" that has not been introduced before.
The blocks have been changed with the addition of new links. Slightly improved goal. Below the goal is a paragraph about fundamentally new research in this direction with links.
MATERIALS AND METHODS: This section should be rewritten by reorganising it so that the experimental sample, the characteristics of the sample and the methodology used for the cognitive and neurophysiological data are precise.
The characteristics of the group are partially included at the end of the paragraph ("Study group consisted of ....).
In addition, the sources of the instruments used (e.g. SCL-90 R, Mini Mult) should be included for validity.
The procedure for the acquisition of electroencephalographic data is unclear, and the motivation for the choice of specific indicators is not explained '....to select the most valid indicators under time constraints... ).
Statistical analysis: the methodology applied needs to be clarified. Furthermore, an algorithm is referred to but needs to be described.
The chapter Methods has been changed in structure, into subchapters (subjects, psychology, neurophysiology, processing) with the addition of references.
RESULTS: should be reported more clearly with reference to the methodology described in the previous paragraph.
The outcomes are structured under methods into subchapters.
DISCUSSION: the discussion is unclear in places, and the periods are unconnected; they should be rewritten with more contextualisation. Understand the results point by point and discuss them concerning the figures created.
The Discussion has been reformatted under a clearer ideology for evaluating, on the one hand, psycho-emotional reactions, and on the other hand, sensory (informational, cognitive) with the addition of new references. According to this scheme, the Introduction, Methods and Results have also been rebuilt.

Round 3
Reviewer 1 Report
Dear Author,
the changes you have made have improved the manuscript. However, some aspects remain to be improved. I will indicate them below:
1. materials and methods: indicate the sub-sections by numbering them e.g. psychological testing 2.1 etc..
2. why was the paragraph with the limitations inserted here? If it only concerns the group it should be inserted within the previous period or otherwise indicated at the end of the paper.
3. indicate sub-section with details of the statistical analysis conducted on the variables considered.
4. as for materials and methods please number the sub-sections
5. table 1: legend missing
6. line 364 ff: statistical tests not previously introduced, please elaborate in materials and methods
7. lines 369-372 (and elsewhere in the results): comments of this kind referring to other publications should be included in the discussion section (in addition to citing the studies with references). I recommend 'cleaning up' the results section from comments and interpretations intended for discussion.
8. table 4: Alpha , Beta1, Beta2 values are not shown, if the analysis was not conducted, remove the columns or insert the values
9. figure 4: not clear: I recommend setting it up better by indicating the frequency bands in the ordinate . Also in the legend you describe the left and the right side, the image on the bottom left what does it refer to?
10. lines 437-449: sorry but this period is not clear. I recommend that the results be rearranged by creating subsections so that they are more usable. Also as stated above, delete everything outside the pure results, comments and interpretations go in the discussion section.
11. figure 6: factor analysis of what? Add the variables considered on the axes
12. figure 7: regression analysis of what? Insert dependent and independent variables on the axes. Also enter the regression line formula or at least the regression values.
13. line 487: which tests? I would also start the discussion by resuming the objective of the study so. E.g. "the objective of the present work was to .... the analysis conducted between xxx generally showed that etcc..." and then move on to an in-depth discussion.
14. in the discussion I recommend referring to the results by citing the relevant figure and/or the significant result.
15. abstract lines 22-23: check word correctness
Author Response
Dear reviewer!
Thanks a lot for your work. Unfortunately, the author does not see the little things. I respond to comments below. Almost everything is taken into account. In the new version, if possible, highlighted in blue.
Sincerely, author
- materials and methods: indicate the sub-sections by numbering them e.g. psychological testing 2.1 etc..
Added
- why was the paragraph with the limitations inserted here? If it only concerns the group it should be inserted within the previous period or otherwise indicated at the end of the paper.
Moved before Conclusion
- indicate sub-section with details of the statistical analysis conducted on the variables considered.
Ok
- as for materials and methods please number the sub-sections
Ok
- table 1: legend missing
Added
- line 364 ff: statistical tests not previously introduced, please elaborate in materials and methods
Added
- lines 369-372 (and elsewhere in the results): comments of this kind referring to other publications should be included in the discussion section (in addition to citing the studies with references). I recommend 'cleaning up' the results section from comments and interpretations intended for discussion.
Removed
- table 4: Alpha , Beta1, Beta2 values are not shown, if the analysis was not conducted, remove the columns or insert the values
Added
- figure 4: not clear: I recommend setting it up better by indicating the frequency bands in the ordinate . Also in the legend you describe the left and the right side, the image on the bottom left what does it refer to?
Added
- lines 437-449: sorry but this period is not clear. I recommend that the results be rearranged by creating subsections so that they are more usable. Also as stated above, delete everything outside the pure results, comments and interpretations go in the discussion section.
Added
- figure 6: factor analysis of what? Add the variables considered on the axes
Added
- figure 7: regression analysis of what? Insert dependent and independent variables on the axes. Also enter the regression line formula or at least the regression values.
Added
- line 487: which tests? I would also start the discussion by resuming the objective of the study so. E.g. "the objective of the present work was to .... the analysis conducted between xxx generally showed that etcc..." and then move on to an in-depth discussion. - let it stay as it is.
- in the discussion I recommend referring to the results by citing the relevant figure and/or the significant result. - added
- abstract lines 22-23: check word correctness - removed